# Influence of Molecular Weight of Polysaccharides from *Laminaria japonica* to LJP-Based Hydrogels: Anti-Inflammatory Activity in the Wound Healing Process

**DOI:** 10.3390/molecules27206915

**Published:** 2022-10-15

**Authors:** Yifan Chen, Weixuan Huang, Yang Chen, Minqian Wu, Ruohan Jia, Lijun You

**Affiliations:** School of Food Science and Engineering, South China University of Technology, Guangzhou 510640, China

**Keywords:** hydrogel, polysaccharides, anti-inflammatory, wound dressing, molecular weight

## Abstract

In this study, polysaccharides from *Laminaria japonica* (LJP) were produced by the treatment of ultraviolet/hydrogen peroxide (UV/H_2_O_2_) degradation into different molecular weights. Then, the degraded LJP were used to prepare LJP/chitosan/PVA hydrogel wound dressings. As the molecular weight of LJP decreased from 315 kDa to 20 kDa, the swelling ratio of the LJP-based hydrogels rose from 14.38 ± 0.60 to 20.47 ± 0.42 folds of the original weight. However, the mechanical properties of LJP-based hydrogels slightly decreased. With the extension of the UV/H_2_O_2_ degradation time, the molecular weight of LJP gradually decreased, and the anti-inflammatory activities of LJP-based hydrogels gradually increased. LJP that were degraded for 60 min (60-gel) showed the best inhibition effects on proinflammatory cytokines, while the contents of TNF-α, IL-6, and IL-1β decreased by 57.33%, 44.80%, and 67.72%, respectively, compared with the Model group. The above results suggested that low Mw LJP-based hydrogels showed great potential for a wound dressing application.

## 1. Introduction

As the largest organ of the human body, the skin can protect the internal environment, resist the invasion of bacteria, and avoid excessive water loss, which is the initial defense of the human body. Numerous health issues, including bacterial infections, protein and moisture loss, immune system diseases, and loss of moisture will develop if the skin is injured and cannot be repaired in time. [1,2,3]. When facing mechanical injury, a complicated wound healing process of four overlapping stages would be triggered in our skin [4]. These four stages are hemostasis [5], inflammatory response [6], cell proliferation, and tissue remodeling [7]. Of those, the inflammatory response stage usually happens several hours after the injury. Inhibition of cell inflammation response is one of the most important means to promote the wound healing process [8].

Traditional wound dressings, such as gauze and cotton, can simply stop the bleeding and protect the wound from secondary pollution. However, they have strong adhesion and will cause pain and discomfort to patients when removed [9,10]. Since the moist healing theory was put forward, hydrogel, a new type of wound dressing with good hydrophilicity and strong absorption ability, has become a research hotspot [11], because hydrogel can keep the wound moderately moist and, therefore, promote the wound healing process [12,13]. It has been reported that polysaccharides can be used to prepare hydrogel wound dressings [14,15]. According to the previous study, algal sulfated polysaccharides usually showed great anti-inflammatory activities by inhibiting the expression of cell proinflammatory factors [16,17]. Therefore, algal sulfated polysaccharides may be used as bioactive substances in wound healing. What is more, the molecular weight of algal sulfated polysaccharides can affect their bioactivities, and play an important role in hydrogel preparation. When algal sulfated polysaccharides were degraded into lower molecular weight, the anti-inflammatory activity significantly increased [18,19]. Polymers with lower molecular weight tend to form hydrogels with higher swelling properties, but the mechanical properties of hydrogels will slightly decrease [20,21,22].

To obtain the algal sulfated polysaccharides with lower molecular weight, various degradation methods have been used to treat polysaccharides. Traditional degradation methods include physical, chemical, and biological approaches. However, they have the disadvantages of high cost, high environmental pollution, and changes of structure, respectively. Our previous studies showed that ultraviolet/hydrogen peroxide (UV/H_2_O_2_) degradation is a novel, non-pollution, and efficient method to degrade polysaccharides [23]. Moreover, UV/H_2_O_2_-degraded algae polysaccharides exhibited significant anti-inflammatory activity.

In our previous study, polysaccharides extracted from *Laminaria japonica* (LJP) were proved to be the most potential algae polysaccharides to form wound dressing hydrogels [24]. Therefore, in this study, LJP were treated with UV/H_2_O_2_ degradation to form polysaccharides with different molecular weights. Moreover, the effects of the molecular weight on anti-inflammatory activity, mechanical properties, and swelling properties of LJP-based hydrogels were investigated.

## 2. Results

### 2.1. Chemical Composition

As shown in Table 1, the contents of protein in LJP ranged from 0.20–1.25%, which showed no regularity with the extension of the degradation time. The contents of reducing sugar and uronic acid gradually increased as the degradation time extended, while the contents of the sulfate group decreased as the degradation time extended because, after UV radiation, H_2_O_2_ transferred into hydroxyl radicals [25]. Part of the polysaccharide chains (for example, C-H groups in carbon 1 and carbon 4) was attacked by these hydroxyl radicals [26]. Then, a rapid degradation occurred, and reductive terminals were produced, which led to an increase in the contents of reducing sugar in LJP. The increased contents of uronic acids in LJP might be explained by the greater exposure of uronic acid groups inside the LJP chains after the UV/H_2_O_2_ degradation treatment [27,28]. However, the decrease in the contents of sulfate groups may be caused by the shedding of sulfate groups during degradation [29].

### 2.2. Molecular Weights

With the extension of the UV/H_2_O_2_ degradation time, the molecular weight (Mw) of LJP decreased gradually from 315.1 kDa at 0 min to 10.5 kDa at 240 min. With the passage of degradation time, the decrease rate of the Mw also showed a downward trend, and it became stable after 60 min. According to the results of the Mw (Figure 1), 0-LJP, 5-LJP, 10-LJP, 30-LJP, and 60-LJP were selected for subsequent hydrogel preparation. The Mw of the above samples was 315.1, 173.7, 79.6, 37.5, and 20.5 kDa, respectively.

### 2.3. Swelling Properties

The swelling process of LJP-based hydrogels (Figure 2A) could be divided into two stages. In the prior stage, the porous structure in the hydrogels allowed them to hold a large number of water molecules inside. Thus, the swelling ratio of the hydrogels increased rapidly in the first 30 min. In the later stage, the swelling rate gradually slowed down and the swelling ratio finally reached an equilibrium of 14–20 folds, because of the spatial constraints of the network structure. According to Figure 2A, the smaller Mw of LJP tended to form hydrogels with a higher swelling ratio. Specifically, the equilibrium swelling ratios of 0-gel, 5-gel, 10-gel, 30-gel, and 60-gel were 14.38 ± 0.60, 15.82 ± 1.62, 17.23 ± 0.63, 18.47 ± 0.33, and 20.47 ± 0.42, respectively. On the one hand, the smaller Mw of LJP was related to shorter molecular chains. This characteristic resulted in lower cross-linking density and allowed more water molecules to enter. On the other hand, the UV/H_2_O_2_ degradation treatment created more -COO- and -OH- groups in LJP. The increase in intermolecular repulsion made the network structure looser [20].

The equilibrium swelling ratios of the Blank group (chitosan-polymer polyvinyl alcohol hydrogel) and the Positive group (gelatin sponge) were 8.49 ± 0.67 and 8.69 ± 0.25, respectively, which were significantly lower than LJP-gels. Traditional hydrogels (Blank and Positive) were single network structures, while LJP-based hydrogels were proved to be double network structures in our previous study [24]. Compared with traditional hydrogels, LJP-gels had obvious advantages in swelling properties because the hydrogel structure with a double network was more stable and more porous, which was more conducive to absorbing water molecules.

### 2.4. Mechanical Properties

According to Figure 2E, a significant decrease was observed in the tensile strength of LJP-gels after UV/H_2_O_2_ degradation. The degradation process broke down polysaccharide chains and formed hydrogels with lower cross-linking density, so the network connection became looser and the tensile strength decreased. The elongation at the break of LJP-gels shown in Figure 2F suggested no significant difference between the Blank group and LJP-gels (*p >* 0.05). The results indicated that the toughness of the hydrogels was mainly provided by the chitosan/polymer polyvinyl alcohol (PVA) skeleton in this hydrogel system. The Young’s modulus shown in Figure 2G claimed that the Positive group owned the highest Young’s modulus, which was considered as a brittle material. However, due to the addition of flexible material PVA, LJP-gels are equipped with both high strength and high toughness. In conclusion, the toughness of LJP-gels was a great breakthrough compared with that of commercial wound dressings. This characteristic allowed LJP-gels to overcome their fragility after absorbing blood. Moreover, a soft LJP-gel dressing with high water content can ease the pain of removing dressings from the wound.

### 2.5. Cell Cytotoxicity

As shown in Figure 2B, cell viabilities of HaCaT cells with LJP treatment all maintained above 90%. The results indicated that LJP with a different Mw (0-LJP, 5-LJP, 10-LJP, 30-LJP, and 60-LJP) did not show any cytotoxicity in the range of 0–1000 μg/mL. After directly contacting with Blank, Positive, 0-gel, 5-gel, 10-gel, 30-gel, and 60-gel for 24 h, the viabilities of HaCaT cells still reached 98.61 ± 0.96%, 99.14 ± 4.32%, 86.34 ± 3.93%, 90.01 ± 4.04%, 84.19 ± 3.18%, 89.61 ± 4.56%, and 88.39 ± 9.91%, respectively, as shown in Figure 2C. These results suggested that the hydrogels were non-cytotoxic and could be used in wound dressings.

### 2.6. Construction of Cell Inflammation Model Analysis

Lipopolysaccharide (LPS), an endotoxin released from Gram-negative bacteria, could induce cell inflammation by activating the secretion of immune-related cytokines [30]. In this study, LPS was used to induce inflammation in HaCaT cells, and the protective effects of LJP and LJP-gels on HaCaT cells were further explored. Different concentrations of LPS were set at 0, 1, 2.5, 5, 10, 20, 40, 80, and 100 μg/mL, respectively. As shown in Figure 3A, with the increase in LPS concentration, the level of proinflammatory cytokines IL-6 showed an increasing trend. When the concentration of LPS reached 80 μg/mL, the content of IL-6 significantly increased to 19.83 ± 2.64 pg/mL, which was 4.25 times higher than that of the Control group (3.78 ± 0.78 pg/mL). The results suggested that 80 μg/mL was the optimum LPS concentration for setting up the HaCaT cell inflammation model.

In the LPS-induced cell inflammation model, the treatment time of LPS was another important affecting factor. After determining the concentration of LPS as 80 μg/mL, the treatment time of LPS was further changed as follows: 0, 2, 4, 6, 8, 12, and 24 h. The secretions of IL-6 in the group without LPS treatment (Control group) and the group with LPS treatment Model group) were measured, and the results are shown in Figure 3B. The results showed that the highest level of IL-6 secretion appeared at 12 h after treatment, which increased significantly from 3.94 ± 0.11 to 20.37 ± 1.23 pg/mL. Therefore, 12 h was the optimum LPS treating time for constructing the HaCaT cell inflammation model.

### 2.7. Effects of LJP and LJP-Gels on Proinflammatory Cytokines and Related Genes Expression

When facing mechanical damage, human tissues would produce proinflammatory cytokines, such as TNF-α, IL-6, and IL-1β, to inhibit the inflammatory reaction by regulating the release of inflammatory mediators, chemokines, and interferons [31]. According to Figure 4A–C, the levels of TNF-α, IL-6, and IL-1β in the Model group increased by 2.97, 3.20, and 4.83 times, respectively, compared with the corresponding Control group. As shown in Figure 4A, 0-LJP and 5-LJP showed no inhibitory effects on cell excretive TNF-α. However, with the extension of the UV/H_2_O_2_ degradation time, the molecular weight of LJP decreased, and the inhibition of 10-LJP, 30-LJP, and 60-LJP on TNF-α gradually increased. Specifically, the contents of TNF-α decreased by 15.42%, 33.86%, and 38.48%, respectively, compared with the Model group. According to Figure 4B, an increasing trend of the inhibitory effects on cell excretive IL-6 was observed as the molecular weight of LJP decreased. Specifically, 0-LJP showed no inhibitory effects on cell excretive IL-6, while the contents of IL-6 in the 5-LJP, 10-LJP, 30-LJP, and 60-LJP treatment groups decreased by 10.78%, 15.14%, 16.98%, and 28.93%, respectively, compared with the Model group. As for IL-1β shown in Figure 4C, there was no significant difference in the inhibitory effects of 0-LJP, 5-LJP, and 10-LJP (*p* > 0.05), while the contents of cell excretive IL-1β decreased by 24.52–27.64%, compared with the Model group. The inhibitory effects of 30-LJP and 60-LJP on cell excretive IL-6 were enhanced, while the contents of IL-1β decreased by 38.07 and 41.87%, respectively, compared with the Model group.

The inhibitory effects of LJP-gels on proinflammatory factors were similar to those of LJP. With the extension of the UV/H_2_O_2_ degradation time, the molecular weight of LJP decreased, and the inhibitory effects of LJP-based hydrogels gradually increased. As shown in Figure 4A, compared with the Model group, the Blank group (chitosan-PVA hydrogel), the Positive group (gelatin sponge), and 0-gel did not show any inhibitory effects on cell excretive TNF-α. However, 5-gel, 10-gel, 30-gel, and 60-gel showed an increased inhibitory effect on cell excretive TNF-α, while the contents of TNF-α decreased by 12.24%, 36.53%, 41.22%, and 57.33%, respectively. According to Figure 4B, 0-gel did not show any inhibitory effects on cell excretive IL-6, while the contents of IL-6 in 5-gel, 10-gel, 30-gel, and 60-gel treatment groups decreased by 31.89%, 38.94%, 42.43%, and 44.80%, respectively, compared with the Model group. Meanwhile, the Blank group showed a high inhibitory effect and could decrease IL-6 secretion by 28.46%, but the Positive group showed no inhibitory effects. As for IL-1β shown in Figure 4C, there was no significant difference in the inhibitory effects of 0-gel and 5-gel (*p* > 0.05), while the contents of cell excretive IL-1β decreased by 43.90% and 49.88%, compared with the Model group. However, the inhibitory effects of 10-gel, 30-gel, and 60-gel on cell excretive IL-1β gradually increased, while the contents of IL-1β decreased by 54.54%, 61.69%, and 67.72%, respectively, compared with the Model group. Meanwhile, the Blank group and the Positive group both showed an inhibition effect, while the contents of IL-1β decreased by 24.98% and 10.52%, respectively.

In conclusion, with the extension of the UV/H_2_O_2_ degradation time, the molecular weight of LJP decreased from 350 kDa to 20 kDa, and the degradation process promoted the anti-inflammation effects of LJP. The lower molecular weight of LJP led to higher water solubility, which allowed LJP to be absorbed easier and, therefore, led to an increase in anti-inflammatory activities [8]. The inhibitory effects of LJP-gels on proinflammatory factors significantly enhanced. LJP-gels had more proinflammatory factor inhibition effects than the Blank group did, indicating that LJP are the main anti-inflammatory active substance in LJP/chitosan/PVA hydrogel system.

Among all the LJP-gels, 60-gel showed the best effects on anti-inflammatory. Thus, the effects of 60-gel on the expression of proinflammatory related genes were further investigated. According to Figure 4D–F, 60-gel could play an anti-inflammatory role by down-regulating the expression of cytokines TNF-α, IL-1β, and IL-6.

## 3. Discussion

Hydrogel is a kind of polymer material with a network structure, in which there is a hydrophilic group. Therefore, it could absorb a large amount of water and combine with water firmly [32]. Compared with traditional dressings, hydrogel could keep wounds moderately moist, absorb exuded tissue fluid, promote wound healing, relieve pain, inhibit bacterial growth, and improve the microenvironment, which is an excellent kind of new wound dressing [33,34]. The materials commonly used to prepare hydrogel are mainly divided into natural polymers and synthetic polymers. Natural polymers, including proteins, peptides, and polysaccharides, have strong hydrophilicity, good biocompatibility, and other advantages [35]. Algae polysaccharides are anionic polysaccharides, which can form ionic cross-linking with chitosan and make the structure more stable. Moreover, chitosan has hemostatic and antibacterial effects, which are suitable for preparing wound dressing hydrogels [36]. Although the hydrogels prepared by common natural polymers have the properties of high safety and good biocompatibility, they also have the problem of low mechanical strength, so they can easily break after blood feeding. To improve the mechanical properties of hydrogels, this study added artificial polymer polyvinyl alcohol (PVA) to natural polymer hydrogels, which is a synthetic polymer commonly used to enhance the mechanical strength of hydrogels. It can be combined with polysaccharides to form hydrogels by freezing and thawing [37].

In this study, polysaccharides from *Laminaria japonica* (LJP) with different molecular weights were prepared by the UV/H_2_O_2_ degradation method. The swelling properties varied significantly depending on the nature of the cross-linker [20]. The swelling ratio of LJP-gels significantly increased after the UV/H_2_O_2_ treatment, because shorter molecular chains would form the structure with lower cross-linking density. When the structure of the hydrogels became looser, the pores became bigger, which would make it more conducive to the entry of water [38].

HaCaT cells are now widely used to investigate the wound healing of the skin [39]. Furthermore, this study discussed the four stages of wound healing. Polysaccharides extracted from brown marine algae represented an obvious ability for platelet aggregation [40]. In the hemostatic stage, LJP-based hydrogels with a large molecular weight had the best blood-sucking capacity and coagulation effect. They could absorb 15 times their weight of blood because of the stable structure, and LJP could promote platelet aggregation, thus promoting blood coagulation. In the inflammatory reaction stage, LJP-based hydrogels with low molecular weight had the best anti-inflammatory effect. Moreover, they could downregulate the expression of *TNF-α*, *IL-1β*, and *IL-6* genes in cells. Our previous studies have shown that algae polysaccharides with a small molecular weight prepared by the UV/H_2_O_2_ degradation method have good anti-inflammatory activity [16]. At the cell proliferation stage, LJP-based hydrogels with a higher molecular weight had the best healing effect. After treating HaCaT cells with mechanical injury with 0-gel for 24 h, the wound can be completely healed. Moreover, PVA-gel also played an important role in mechanical properties. The glutaraldehyde cross-linked PVA hydrogels exhibited high mechanical strength, which could be attributed to the formation of hydrogen bonds [41]. Collagen, the principal structural component of the extracellular matrix in the dermis, is responsible for the strength and resilience of the skin. The expression of collagen could contribute to promoting skin tissue organization and repair [42]. In the stage of tissue remodeling, LJP-based hydrogels with lower molecular weight represented the best self-repairing effect, which can promote the release of hydroxyproline and the production of collagen in HaCaT cells, and increase the relative expression of *COAI* gene.

Therefore, this study aimed to prepare a hydrogel-type wound dressing with a promoting healing effect by using algae polysaccharides with various biological activities. Additionally, this study aimed to investigate how hydrogels affected the four stages of wound healing—hemostasis, inflammatory response, cell proliferation, and tissue remodeling—by reducing inflammation. The results of this study will pave the way for the preparation and application of algae polysaccharides-based hydrogels with anti-inflammatory functions.

## 4. Materials and Methods

### 4.1. Materials and Chemicals

*Laminaria japonica* were obtained from Fuzhou, China. Modified Eagle’s medium (MEM), fetal bovine serum (FBS), insulin, penicillin, and streptomycin were purchased from Gibco Co. (Carlsbad, CA, USA). LPS and Dextran standards were purchased from Sigma-Aldrich Chemical Co. (St. Louis, MO, USA). Chitosan (degree of deacetylation = 90%) was purchased from Shanghai Yuanye Bio-Technology Co., Ltd. (Shanghai, China). Human immortalized keratinocytes (HaCaT cells) were obtained from Cell Resource Center, Shanghai Institute of Biological Sciences (Shanghai, China). MTT kit was purchased from Nanjing Jiancheng Bio-Technology Co., Ltd. (Nanjing, China). ELISA kits of IL-1β, IL-6, and TNF-α were obtained from MultiSciences Biotech, Co., Ltd. (Hangzhou, China).

### 4.2. Preparation and Degradation of LJP

Polysaccharides from *Laminaria japonica* (named LJP) were extracted by the hot water extraction method according to our previous report [24]. Then, to obtain LJP with different molecular weights, they were prepared from *Laminaria japonica* according to our previously reported method with some modifications [16]. Specifically, 25 mL LJP solution (3 mg/mL) was mixed with 5 mL H_2_O_2_ solution (60 mmol/L) in a 90 mm glass culture dish and placed in a UV irradiator (HOPE-MED 8140 equipped with a UV lamp model of UVB-313EL and wavelength of 313 nm, Tianjin Hepu Company, Tianjin, China) to irradiate for different time periods (0, 5, 10, 30, and 60 min). The reaction was stopped after the UV treatment and manganese dioxide was used to remove the residual H_2_O_2_. Then, the degraded LJP solution was collected and concentrated (60 °C, 1/10 of the original volume), centrifuged (8000 rpm, 15 min), dialyzed (4 °C, 3000 Da, 48 h), and lyophilized (−40 °C, 48 h). The degraded LJP were 0-LJP, 5-LJP, 10-LJP, 30-LJP, and 60-LJP, respectively.

### 4.3. Characterization of LJP

#### 4.3.1. Chemical Composition

The contents of sulfate groups, uronic acids, reducing carbohydrates, and proteins of LJP were determined by the barium sulfate turbidimetry method [43], the sulfuric acid-carbazole method [44], the 3,5-dinitrosalicylic acid method [45], and the Coomassie bright blue colorimetry method [46], respectively. The standards for quantification of the mentioned tests were potassium sulfate, galacturonic acid, glucose, and bovine serum albumin, respectively.

#### 4.3.2. Molecular Weight

The molecular weight of the degraded LJP was determined by high-performance gel-permeation chromatography (Model 2414, Waters Co., Milford, MA, USA) as previously described [23]. A TSK G6000 PWXL column (7.8 × 300 mm i.d., 13 μm, Tosoh Co., Ltd., Tokyo, Japan) and a TSK G3000 PWXL column (7.8 × 300 mm i.d., 7 μm, Tosoh Co., Ltd.) were connected in series. Dextran with molecular weights of 4.320, 12.6, 126, 289, and 496 kDa, respectively, were used as the standards. Both the standards and samples (30 μL each run) were eluted with 0.02 M monopotassium phosphate (0.5 mL/min) at 35 ± 0.1 °C.

### 4.4. Preparation of LJP-Based Hydrogels

The LJP-based hydrogels were prepared according to the previous report [24] with the same volume of the following solutions: 0.5% (*w/w*) LJP solution, 1% (*w/w*) chitosan solution (adjust pH value to 3.0 with 6 M hydrochloric acid), and 5% (w/w) polyvinyl alcohol solution (bath in 70 °C). Chitosan solution was cross-linked with LJP by 5-min continuous stirring. The LJP-chitosan mixture was dropwise added into the PVA solution. After the repeated freeze-thawing process, the LJP-based hydrogels were formed. The hydrogels made up of 0-LJP, 5-LJP, 10-LJP, 30-LJP, and 60-LJP were named 0-gel, 5-gel, 10-gel, 30-gel, and 60-gel, respectively.

### 4.5. Group of Experiments

Five LJP-based hydrogels (0-gel, 5-gel, 10-gel, 30-gel, and 60-gel) were included in the sample groups. Gelatin sponge (named Positive) was selected as the positive control group. Chitosan-PVA hydrogel (without adding LJP in the hydrogel system) was deemed as the blank control group (named Blank).

### 4.6. Swelling Properties

Lyophilized hydrogels (roughly 20 mg) were soaked in distilled water and weighed every 30 min after wiping off any extra moisture. The swelling ratio (*SR*) was obtained by Formula (1), where “*W_t_*” represents the weight of the hydrogels at *t* min, and “*W_0_*” represents the dry weight of the hydrogels.
(1)$$SR=\frac{W_{t}-W_{0}}{W_{t}}\;$$

### 4.7. Mechanical Properties

Wet hydrogels were shaped into 10 × 60 mm with a thickness of 2–4 mm, and their mechanical properties were determined by uniaxial tensile tests using a universal material testing machine (INSTRON5565, Instron Ltd., Boston, MA, USA). The clamping intercept was set as 40 mm and the rate of extension was set as 50 mm/min. Stress (MPa)-strain (%) curve was attained, tensile strength (*TS*), and elongation at break (*E*), as well as Young’s modulus (*Y*), were calculated by the Formulas (2)–(4),
(2)$$TS=\frac{F_{\max}}{A}$$
where *F*_max_ was the maximum stress that the hydrogels could bear, and *A* was the original cross-sectional area of the hydrogels, and
(3)$$E\left(\%\right)=\frac{L}{L_{0}}\times 100\%$$
where *L* represents the corresponding strain of *F*_max_, and *L*_0_ represents the original length, which was equal to the clamping intercept.
(4)$$Y=\frac{\sigma_{90\%}-\sigma_{30\%}}{\varepsilon_{90\%}-\varepsilon_{30\%}}$$

The linear ranges of the stress-strain curves were selected in order to calculate Young’s modulus. According to the stress-strain curves, data points with tensile strains of 90% and 30% were selected, where *σ*_30%_ and *σ*_90%_ were the stress at 30% and 90%, respectively, and where *ε*_30%_ and *ε*_90%_ were the strain at 30% and 90%, respectively.

### 4.8. Cell Culture

Human Keratinocytes cells (HaCaT) were cultured in MEM, supplemented with 10% FBS and 1% penicillin-streptomycin antibodies at 37 °C with an atmosphere of 5% CO_2_. The medium should be changed every two days and the growth situation of cells should be observed on time.

#### 4.8.1. Pretreatments of LJP and LJP-Gels

Lyophilized hydrogels (0-gel, 5-gel, 10-gel, 30-gel, 60-gel, Blank, and Positive) were shaped in suitable sizes and were pre-swelled with phosphate buffer solution (PBS) for 30 min. Subsequently, the swelled hydrogels were soaked in 75% ethanol for 30-min sterilization. After the ethanol was removed, the hydrogels were then soaked into PBS to remove any residual ethanol. The PBS was changed every 30 min until the residual ethyl alcohol was removed.

LJP were dissolved in MEM and filtrated by a 0.22 μm filter. The LJP solution was diluted into the same concentrations as in the LJP-based hydrogel systems.

#### 4.8.2. Cell Cytotoxicity in HaCaT

HaCaT cells were seeded in 96-well plates with a density of 104/well. Each 96-well plate was equipped with a Blank control group and a Control group. Wells without cells acted as the Blank control group and wells with cells but without drug exposure served as the Control group. When HaCaT cells proliferated to 80%–90% of the well area, LJP with different molecular weights (0-LJP, 5-LJP, 10-LJP, 30-LJP, and 60-LJP) and pre-sterilized LJP-gels (0-gel, 5-gel, 10-gel, 30-gel, 60-gel, Blank, Positive) were severally added. After 24h of drug treatments, the cell cytotoxicity was determined using the MTT kit under 570 nm with a microplate reader (SpectraMax190, Molecular Devices, Sunnyvale, CA, USA).

#### 4.8.3. Construction of Cell Inflammation Model

HaCaT cells were seeded in 12-well plates with a density of 10^5^/well. To explore the optimum LPS concentration, lipopolysaccharide (LPS) with different concentrations (0, 1, 2.5, 5, 10, 20, 40, 80, and 100 μg/mL) was added when HaCaT cells proliferated to 80%–90%. After 24 h of LPS treatments, the supernatant was collected to determine the contents of IL-6 using the IL-6 ELISA kit.

To explore the optimum LPS treating time, 80 μg/mL of LPS was added when HaCaT cells proliferated to 80%–90%. After different durations of LPS treatment (0, 2, 4, 6, 8, 12, and 24 h), the supernatant was collected to determine the contents of IL-6 using the IL-6 ELISA kit.

#### 4.8.4. Inflammatory Cytokines Measurement

HaCaT cells were seeded in 12-well plates with a density of 10^5^/well. Each 12-well plate was equipped with a control group and a model group. Wells without LPS nor drug exposure acted as the Control group, and wells with LPS but without drug exposure acted as the Model group. When HaCaT cells proliferated to 80%–90% of the well area, 250 μg/mL of LJP and 20 mg of presterilized LJP-gels were added. After 12 h of drug treatments, 80 μg/mL of LPS was added for another 12 h to induce cell inflammation and the supernatant was collected. The contents of TNF-α, IL-6, and IL-1β were measured using ELISA kits according to the manufacturer’s protocol.

#### 4.8.5. qRT-PCR Analysis

HaCaT cells were seeded in 12-well plates and treated with the methods described in Section 4.8.4. The supernatant was abandoned and the RNA inside the adherent cells was extracted with 1 mL Trizol regent in each well. Then, the concentration of extracted RNA was determined under 260 nm using a micro-spectrophotometer (K5800C, Beijing Kaiao Technology Development Co., Ltd., Beijing, China). According to the quantitative results, the RNA samples were diluted to 2 μg/mL, and 1 μL of diluted RNA was transferred to cDNA according to the methods recommended by RevertAid First Strand cDNA Synthesis Kit (K1622, Applied Biosystems, Foster City, CA, USA) with oligo d(T) primers. The expression levels of human proinflammatory-factor-related genes TNF-α, IL-6, and IL-1β were determined according to the methods recommended by the SYBR^®^ Select Master Mix (4472908, Invitrogen, Carlsbad, CA, USA) using a Mini Opticon Real-Time PCR System (Bio-Rad, Hercules, CA, USA). The Human *GAPDH* gene was used as an internal reference gene, and the 2^−^^ΔΔCt^ method was used to calculate the fold change (Cq) of the expression level of relative genes. The qRT-PCR primers used are shown in Table 2. The processes of gene amplification were as follows: incubation holding at 95 °C for 10 min, then 40 cycles were performed at 95 °C for 15 s and 60 °C for 1 min.

### 4.9. Statistical Analysis

The data were shown as Mean ± SD (n = 3) and were analyzed with OriginPro 2017 (Origin software, OriginLab, Northampton, MA, USA). Significant differences were analyzed by the Student *t*-test and expressed by a, b, c, d (*p* < 0.05).

## 5. Conclusions

Polysaccharides from *Laminaria japonica* were degraded using the UV/H_2_O_2_ method and LJP/chitosan/PVA hydrogels were prepared with LJP of different molecular weights. The relationship between the molecular weights of LJP and the effects of LJP-based hydrogels on the inflammatory reaction stage in the wound healing process was investigated. After treatment with UV/H_2_O_2_, both LJP and LJP-gels showed enhanced anti-inflammatory effects on HaCaT cells. With the extension of the UV/H_2_O_2_ degradation time, the molecular weight of LJP gradually decreased, and the anti-inflammatory activities of LJP-based hydrogels gradually increased. Therefore, the UV/H_2_O_2_ degradation method was a practical method to improve LJP-based hydrogels’ anti-inflammatory effect on the wound healing process. In addition, low Mw LJP-based hydrogels showed great potential for application in wound dressings.

## Figures and Tables

**Figure 1 molecules-27-06915-f001:**
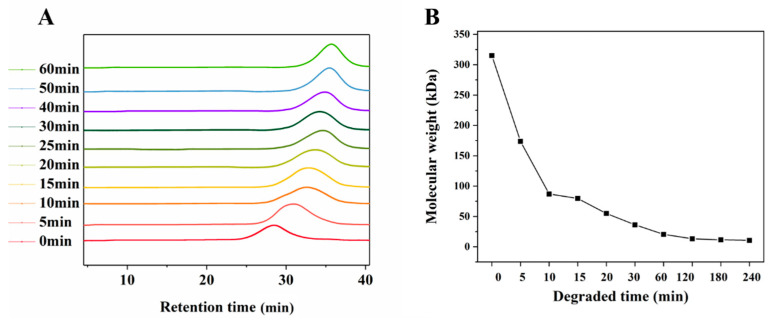
GPC profile (**A**) and changes of molecular weights (**B**) of degraded LJP.

**Figure 2 molecules-27-06915-f002:**
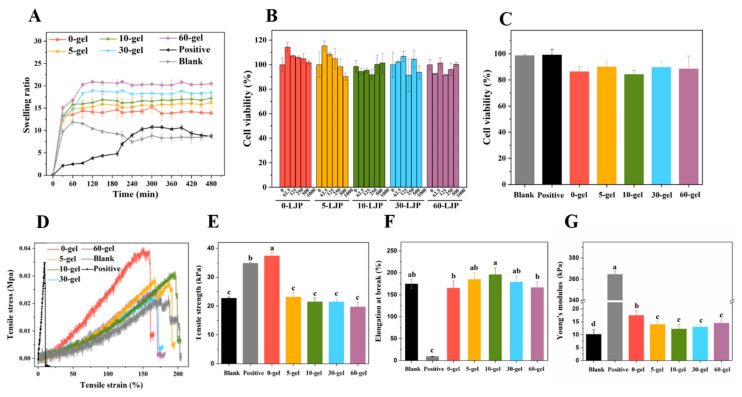
The swelling properties of LJP-based hydrogels (**A**). The cell viabilities of LJP (**B**) and LJP-based hydrogels (**C**). The stress-strain curves (**D**), tensile strength (**E**), elongation at break (**F**), and Young’s modulus (**G**) of LJP-based hydrogels. ^a–d^ Values with different letters are significantly different (*p* < 0.05).

**Figure 3 molecules-27-06915-f003:**
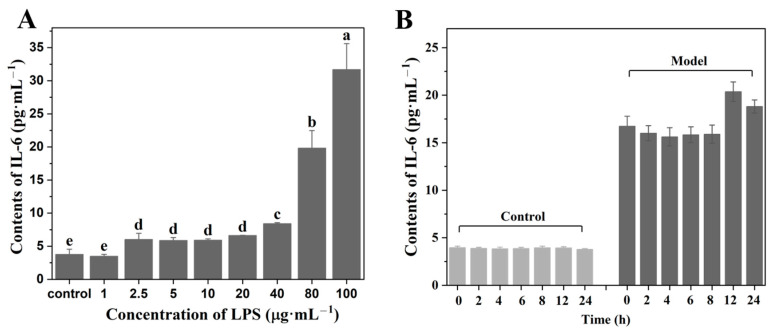
Contents of IL-6 on HaCaT cells of different concentrations of LPS treatment (**A**) and different durations of LPS treatment (**B**). ^a–^^e^ Values with different letters are significantly different (*p* < 0.05).

**Figure 4 molecules-27-06915-f004:**
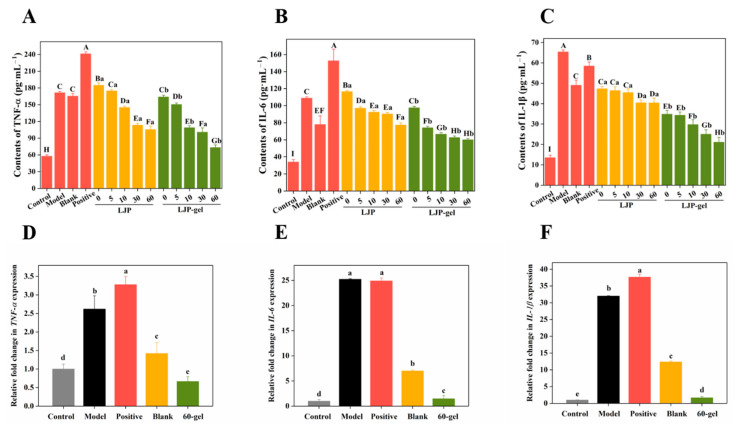
The levels of TNF-α (**A**), IL-6 (**B**), and IL-1β (**C**) secreted by HaCaT cells after LJP and LJP-gels treatment. The gene expressions of TNF-α (**D**), IL-6 (**E**), and IL-1β (**F**) in HaCaT cells after LJP-gels treatment. ^A–I^ Values with different letters are significantly different (*p* < 0.05). ^a–^^e^ Values represent the significant difference between LJP and LJP-gel at the same degradation time (*p* < 0.05).

**Table 1 molecules-27-06915-t001:** Chemical components of LJP.

Composition	Reducing Sugar (%)	Sulfate Group (%)	Uronic Acids (%)	Protein (%)
0-LJP	6.93 ± 0.21 ^c^	7.89 ± 0.11 ^a^	30.31 ± 0.15 ^e^	1.46 ± 0.03 ^b^
5-LJP	7.85 ± 0.41 ^b^	6.95 ± 0.04 ^b^	33.83 ± 0.36 ^d^	0.65 ± 0.03 ^c^
10-LJP	7.92 ± 0.35 ^b^	6.67 ± 0.08 ^c^	36.85 ± 0.21 ^c^	1.84 ± 0.01 ^a^
30-LJP	8.18 ± 0.47 ^a^	6.50 ± 0.10 ^c^	43.34 ± 0.65 ^b^	0.72 ± 0.03 ^c^
60-LJP	8.37 ± 0.27 ^a^	6.35 ± 0.02 ^d^	47.56 ± 0.48 ^a^	0.21 ± 0.02 ^d^

Data are expressed as means ± SDs (n = 3). ^a–e^ Values with different superscripted letters in the same column are significantly different (*p* < 0.05).

**Table 2 molecules-27-06915-t002:** The primers sequences in RT-qPCR.

Gene	Primers Sequences (5′-3′)		Amplicon Size (bp)
GAPDH	Forward	CCCTCTGGAAAGCTGTGG	220
	Reverse	GCTTCACCACCTTCTTGATGT	
TNF-α	Forward	GCTGCACTTTGGAGTGATCG	112
	Reverse	CTTGTCACTCGGGGTTCGAG	
IL-6	Forward	CTGACCCAACCACAAATGC	162
	Reverse	TCTGAGGTGCCCATGCTAC	
IL-1β	Forward	CTGTACCTGTCCTGCGTGTT	199
	Reverse	AGACGGGCATGTTTTCTGCT	

## Data Availability

Not applicable.

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
