# Peer review of "Influence of Molecular Weight of Polysaccharides from Laminaria japonica to LJP-Based Hydrogels: Anti-Inflammatory Activity in the Wound Healing Process"

_molecules, 2022, doi:10.3390/molecules27206915_

Round 1

Reviewer 1 Report

The paper by Zhang et. al prepared different molecular weights of LJP and LJP/ chitosan /PVA hydrogels with UV/H2O2 method, and both LJP and LJP-gels showed enhanced anti-inflammatory effects on HaCaT cells after treatment with UV/H2O2.They found that low Mw LJP-based hydrogels showed great potential for application in wound dressings.

On the whole, authors are basically qualified for the completeness and logical completion of the paper.

In my opinions, authors should address the following comments, in order to make this work suitable for publication in Molecules.

1Table1 showed the chemical components of LJP, Why did reducing sugar , sulfate group and uronic acids change regularly, but not total sugar and protein content? Whether it has been tested multiple times? Whether there is test contingency?

2In Result 2.6 (Line 140-144), why was 80 μg/mL the best concentration? Since you chose 12 hours as the highest amount of IL-6 for the treatment time, why did not you choose 80 μg/mL for the concentration?

3There are something wrong with your figures, please modify it. The specific problems are as follows:

(1) Figure1and 2 didn't explain small figures such as ABCD;

(2) Figure 2 was barely visible, please replace the picture with a clearer one;

(3) There was no error line in figure 2C;

(4) Some of the letters were not legible in figure 4;

(5) The size and thickness of the letters on the abscissa were different in figure 3.

4There are some grammar mistakes in your article. Please correct them.

(1) Line 237-238:“because shorter molecular chains would form a structure with lower cross-linking density, so the structure of the hydrogel became loose.

(2) Line 276: "The reactions were terminated after UV irradiation was completed”

(3) Line 304-306:The hydrogels made from 0-LJP, 5-LJP, 10-LJP, 30-LJP, and 60-LJP were named 0-gel, 5-del, 10- 305 gel, 30-gel, and 60-gel, respectively.

(4) There is a contradiction between line 53 “LJP were” and line 201 “LJP was”. Similar situations occur elsewhere in the article. Please correct them.

Author Response

Thank you for helping us to improve our manuscript. Please find attached revised manuscript entitled ‘Influence of molecular weight of polysaccharides from Laminaria japonica to LJP-based hydrogels: anti-inflammatory activity in the wound healing process’, as well as detailed point-by-point responses to the reviewer’s comments (see below).

We much appreciate the reviewer’s contribution and scientific merits of our manuscript. We have revised the manuscript according to their suggestions.

Please contact me should you have any questions or need additional information. We are looking forward to your positive response.

1、Table1 showed the chemical components of LJP, Why did reducing sugar , sulfate group and uronic acids change regularly, but not total sugar and protein content? Whether it has been tested multiple times? Whether there is test contingency?

Response: Thank you so much for your comments. We have tested the chemical components for several times. However, there was no regularity between degradation time and the contents of total carbohydrate or protein. We think that the differences in data caused by fluctuations may be due to the extremely low content of protein. As for the total carbohydrate, we decided to delete the data of this part.

2、In Result 2.6 (Line 140-144), why was 80 μg/mL the best concentration? Since you chose 12 hours as the highest amount of IL-6 for the treatment time, why did not you choose 80 μg/mL for the concentration?

Response: Thank you so much for your comments. When the concentration of LPS reached 80 μg/mL, the content of IL-6 significantly increased to 19.83±2.64 pg/mL compared with 40 μg/mL LPS. When the concentration of LPS reached 100 μg/mL, the content of IL-6 increased to 31.70±3.91 pg/mL, which may indicate an excessive injury in HaCaT cells. After determining the concentration of LPS as 80 μg/mL, the treatment time of LPS was further decided. Based on the contents of IL-6, we chose 12 hours for the treatment time (the IL-6 content is 20.37±1.03 pg/mL), and there was no significant difference between 12 h treatment and 80 μg/mL LPS (the IL-6 content is 19.83±2.64 pg/mL)

3、There are something wrong with your figures, please modify it. The specific problems are as follows:

(1) Figure 1 and 2 didn't explain small figures such as ABCD;

(2) Figure 2 was barely visible, please replace the picture with a clearer one;

(3) There was no error line in figure 2C;

(4) Some of the letters were not legible in figure 4;

(5) The size and thickness of the letters on the abscissa were different in figure 3.

Response: Sorry for our mistakes. In Fig.1 and Fig.2, a-d values with different letters are significantly different (p < 0.05). Due to the resolution, actually, there were error lines in figure 2C, and we have replaced clearer pictures for all figures. In addition, we have adjusted the size and thickness of the letters on the abscissa in Fig.3.

4、There are some grammar mistakes in your article. Please correct them.

(1) Line 237-238:“because shorter molecular chains would form a structure with lower cross-linking density, so the structure of the hydrogel became loose”.

(2) Line 276: "The reactions were terminated after UV irradiation was completed”

(3) Line 304-306: “The hydrogels made from 0-LJP, 5-LJP, 10-LJP, 30-LJP, and 60-LJP were named 0-gel, 5-gel, 10-gel, 30-gel, and 60-gel, respectively”.

(4) There is a contradiction between line 53 “LJP were” and line 201 “LJP was”. Similar situations occur elsewhere in the article. Please correct them.

Response: Thank you for your attention to those mistakes. (1) We have replaced the sentence “because shorter molecular chains would form a structure with lower cross-linking density, so the structure of the hydrogel became loose” with the sentence “because shorter molecular chains would form the structure with lower cross-linking density. When the structure of hydrogel became looser, the pores became bigger, which would be more conducive to the entry of water”. (2) We have replaced the sentence “The reactions were terminated after UV irradiation was completed” with the sentence “The reaction was stopped after UV treatment”. (3) We have replaced the sentence “The hydrogels made from 0-LJP, 5-LJP, 10-LJP, 30-LJP, and 60-LJP were named 0-gel, 5-gel, 10-gel, 30-gel, and 60-gel, respectively” with the sentence “The hydrogels made of 0-LJP, 5-LJP, 10-LJP, 30-LJP, and 60-LJP were named 0-gel, 5-del, 10-gel, 30-gel, and 60-gel, respectively”. (4) LJP represented polysaccharides from Laminaria japonica. We have replaced all the “LJP was…” with “LJP were…”

Reviewer 2 Report

After a careful review of this MS, here are my comments

·         The introduction was concise, but why was UV/H2O2 selected for degradation in this study?

·         State explicitly what UV/H2O2 method is. Common name for this reaction?

·         In abstract, L16, What is 60-gel? There hasn’t been any mention of gel types. 60 could be LJP or chitosan or PVA content. I believe the abstract can be rewritten for more clarity.

·         In Figures axis labels, units should be in parenthesis

·         Generally, figures are of poor resolution, especially in Figures 2, and 4.
Fig 4A, B, and C. what do the upper and lower case represent? How was mean separated? Include proper captions

·         mutatis mutandis.. is this appropriate in scientific writing?

·         The work lacks originality. Since degradation was carried out, LJP needs to be characterized as in your previous study (Huang et al., 2022). https://doi.org/10.1016/j.algal.2022.102740}. The rheology of hydrogels is an essential parameter as well. This was not done too. Moreso the discussion is superficial and not informative enough.

·         I will suggest a critical rework of the MS.

Author Response

Thank you for helping us to improve our manuscript. Please find attached revised manuscript entitled ‘Influence of molecular weight of polysaccharides from Laminaria japonica to LJP-based hydrogels: anti-inflammatory activity in the wound healing process’, as well as detailed point-by-point responses to the reviewer’s comments (see below).

We much appreciate the reviewer’s contribution and scientific merits of our manuscript. We have revised the manuscript according to their suggestions.

Please contact me should you have any questions or need additional information. We are looking forward to your positive response.

The reviewer tried to carefully read the manuscript titled 'Influence of molecular weight of polysaccharides from Laminaria japonica to LJP-based hydrogels: anti-inflammatory activity in the wound healing process'. Although the content of this paper is not unscientific and unreliable, the reviewer chose 'Major revisions' for this paper. The reviewer has lots of questions and comments on the content as follows.

1、The introduction was concise, but why was UV/H2O2 selected for degradation in this study?

2、State explicitly what UV/H2O2 method is. Common name for this reaction?

Response: Thank you so much for your constructive suggestion. The full name of UV/H2O2 methods is “ultraviolet/ hydrogen peroxide method”, which is used for the degradation of the polysaccharides (some references are as follows).

Actually, Various degradation methods have been used to treat algae polysaccharides, including physical, chemical, and biological approaches. Physical degradation methods, such as ultrasound, microwave, irradiation, and pulsed electric field require expensive equipment, limiting their industrial applications. Chemical degradation methods, such as acid or alkaline hydrolysis have the disadvantages of high energy consumption and severe environmental pollution due to the use of chemical reagents. Biological degradation methods, including enzymatic hydrolysis and microbial fermentation, are mild, rapid, and nontoxic, but searching for specific enzymes and microbes is also tricky because of the complicated structures of algae polysaccharides. Therefore, it is necessary to develop an efficient, feasible, and environment-friendly approach to degrade marine algae polysaccharides. Our previous studies showed that UV/H2O2 degradation is a novel and promising method to degrade marine algae polysaccharides with the advantages of availability, easy operation, mild condition, and without pollution [1-3]. Moreover, UV/H2O2-degraded algae polysaccharides exhibited significant anti-inflammatory activity. We have added the introduction of the method in the introduction part, line 52-59.

Reference

  1. Chen, X.; Zhang, R.; Li, Y.; Li, X.; You, L.; Kulikouskaya, V.; Hileuskaya, K. Degradation of polysaccharides from Sargassum fusiforme using UV/H2O2 and its effects on structural characteristics. Carbohyd. Polym. 2020, 230.
  2. Chen, X.; Sun-Waterhouse, D.; Yao, W.; Li, X.; Zhao, M.; You, L. Free radical-mediated degradation of polysaccharides: Mechanism of free radical formation and degradation, influence factors and product properties. Food Chem. 2021, 365.
  3. Chen, X.; You, L.; Ma, Y.; Zhao, Z.; Kulikouskaya, V. Influence of UV/H2O2 treatment on polysaccharides from Sargassum fusiforme: Physicochemical properties and RAW 264.7 cells responses. Food Chem. Toxicol. 2021, 153, 112246.

3、In abstract, L16, What is 60-gel? There hasn’t been any mention of gel types. 60 could be LJP or chitosan or PVA content. I believe the abstract can be rewritten for more clarity.

Response: Thank you so much for your constructive suggestion. 60-gel referred to the LJP that were degraded for 60 min, and we have added the explanation to the abstract, line 16.

4、In Figures axis labels, units should be in parenthesis

Response: Sorry for our mistakes. We have changed all the figures.

5、Generally, figures are of poor resolution, especially in Figures 2, and 4.

Fig 4A, B, and C. what do the upper and lower case represent? How was mean separated? Include proper captions

Response: Sorry for our mistakes. Small figures with the same letter are not significant at a p-level value < 0.05. In Fig.4, A–I values with different letters are significantly different (p < 0.05). a–b Values represent the significant difference between LJP and LJP-gel at the same degradation time (p < 0.05). We have replaced clearer pictures for all figures.

6、mutatis mutandis.. is this appropriate in scientific writing?

Response: Thank you for your attention to the mistakes. We have replaced the phrase “mutatis mutandis” with the sentence “they were prepared from Laminaria japonica according to our previously reported method with some modifications”.

7、The work lacks originality. Since degradation was carried out, LJP needs to be characterized as in your previous study (Huang et al., 2022). https://doi.org/10.1016/j.algal.2022.102740}. The rheology of hydrogels is an essential parameter as well. This was not done too. Moreso the discussion is superficial and not informative enough.

Response: Thank you so much for your constructive suggestion. In this study, we focused on the influence of molecular weight of LJP to LJP-based hydrogels, and their anti-inflammatory activities in the wound healing process. Hence, we used the UV/H2O2 method to degrade the polysaccharides and explore the anti-inflammatory activities in the four stages of wound healing process. For the characteristic of LJP, we have reported in our previous studies that the shape of the polysaccharides mainly presented non-porous and irregular flaky, and their size became smaller and more uniform with the increase of treatment time [3]. The mechanism of the UV/H2O2 degradation remains to be further studied. In the future, we’ll try to discuss the mechanism. In addition, we have supplemented some discussions to support our results. As for the rheology of hydrogels, we really need to improve in this respect. We are not able to carry out this part of experiments due to the lack of the samples and influence of the epidemic situation.

Reference

  1. Chen, X.; You, L.; Ma, Y.; Zhao, Z.; Kulikouskaya, V. Influence of UV/H2O2 treatment on polysaccharides from Sargassum fusiforme: Physicochemical properties and RAW 264.7 cells responses. Food Chem. Toxicol. 2021, 153, 112246.

Round 2

Reviewer 1 Report

Please provide short descriptions of Figure1A and B, Figure 2A-G, Figure 4A-F in their Figure captions. For example, please indicate what are the figures A and B in the caption of Figure 1. 

Laminaria japonica should be typed in Italic. 

Author Response

Dear Editor,

Thank you again for helping us to improve our manuscript. Please find attached revised manuscript entitled ‘Influence of molecular weight of polysaccharides from Laminaria japonica to LJP-based hydrogels: anti-inflammatory activity in the wound healing process’, as well as detailed point-by-point responses to the reviewer’s comments (see below).

We much appreciate the reviewer’s contribution and scientific merits of our manuscript. We have revised the manuscript according to their suggestions.

Please contact me should you have any questions or need additional information. We are looking forward to your positive response.

1、Please provide short descriptions of Figure1A and B, Figure 2A-G, Figure 4A-F in their Figure captions. For example, please indicate what are the figures A and B in the caption of Figure 1.

Response: Sorry for our mistakes. We have provided the descriptions of Fig.1 (Line 92), Fig.2 (Line 116-118), and Fig.4 (Line 222-224).

2、Laminaria japonica should be typed in Italic.

Response: Thank you so much for your comments. “Laminaria japonica” was mentioned 8 times in the manuscript and we have confirmed that they were typed in Italic. We think that the format may be different due to the version of the word file.

Reviewer 2 Report

The authors have carefully revised this manuscript and satisfactory addressed the concerns of the reviewer. 

Author Response

Thank you again for helping us to improve our manuscript.